# Advances and Challenges in Gene Therapy for Neurodegenerative Diseases: A Systematic Review

**DOI:** 10.3390/ijms252312485

**Published:** 2024-11-21

**Authors:** Nerea García-González, Jaime Gonçalves-Sánchez, Ricardo Gómez-Nieto, Jesús M. Gonçalves-Estella, Dolores E. López

**Affiliations:** 1Institute of Neuroscience of Castilla y León, 37007 Salamanca, Spain; ggmeilan@usal.es (N.G.-G.); jaimegs@usal.es (J.G.-S.); richard@usal.es (R.G.-N.); 2Department of Cellular Biology and Pathology, School of Medicine, University of Salamanca, 37007 Salamanca, Spain; 3Institute for Biomedical Research of Salamanca (IBSAL), 37007 Salamanca, Spain; jgoncalves@usal.es; 4Department of Surgery, School of Medicine, University of Salamanca, 37007 Salamanca, Spain

**Keywords:** Alzheimer’s disease, amyotrophic lateral sclerosis, curative genetic therapy, disease-modifying treatment, Huntington’s disease, Parkinson’s disease, spinal muscular atrophy

## Abstract

This review explores recent advancements in gene therapy as a potential treatment for neurodegenerative diseases, focusing on intervention mechanisms, administration routes, and associated limitations. Following the PRISMA procedure guidelines, we systematically analyzed studies published since 2020 using the PICO framework to derive reliable conclusions. The efficacy of various gene therapies was evaluated for Parkinson’s disease (n = 12), spinal muscular atrophy (n = 8), Huntington’s disease (n = 3), Alzheimer’s disease (n = 3), and amyotrophic lateral sclerosis (n = 6). For each condition, we assessed the therapeutic approach, curative or disease-modifying potential, delivery methods, advantages, drawbacks, and side effects. Results indicate that gene therapies targeting specific genes are particularly effective in monogenic disorders, with promising clinical outcomes expected in the near future. In contrast, in polygenic diseases, therapies primarily aim to promote cell survival. A major challenge remains: the translation of animal model success to human clinical application. Additionally, while intracerebral delivery methods enhance therapeutic efficacy, they are highly invasive. Despite these hurdles, gene therapy represents a promising frontier in the treatment of neurodegenerative diseases, underscoring the need for continued research to refine and personalize treatments for each condition.

## 1. Introduction

Neurodegenerative diseases (NDDs) are a group of disorders affecting the nervous system, marked by their high prevalence and significant impact on patient health [1]. These diseases typically lack curative treatment and have a poor prognosis, leading to progressive disability and reduced life expectancy [2]. NDDs with a genetic basis can be broadly classified into two categories [3], described below.

### 1.1. Monogenic Diseases

Monogenic neurodegenerative diseases are caused by mutations in a single gene, leading to a clear, predictable inheritance pattern and providing a more straightforward target for gene therapy interventions. In contrast to polygenic diseases, where multiple genetic and environmental factors contribute to disease onset, monogenic disorders offer unique opportunities for therapeutic strategies that directly address the underlying genetic defect. Among the most studied monogenic neurodegenerative diseases are Huntington’s disease (HD) and spinal muscular atrophy (SMA), both of which have well-characterized genetic causes. These disorders were chosen as focal points for this review due to their devastating clinical impact, the availability of established animal models, and recent advances in gene therapies [4,5,6,7,8,9,10,11,12,13,14,15].

-Huntington’s disease (HD) is a hereditary degenerative disorder characterized by the expansion of CAG triplet repeats (>36) in the first exon of the *HTT* gene located on chromosome 4q. The repetition of amino acid glutamine causes the production of an abnormal huntingtin (HTT) protein [6]. This protein aggregates causing the extensive degeneration of the cortex and basal ganglia [4,16], with the severity of symptoms correlating with the number of glutamine repeats [17].-Spinal muscular atrophy (SMA) is an autosomal recessive neurodegenerative disease caused by a biallelic mutation of the neuronal survival gene *SMN1* on chromosome 5q. It is the main cause of childhood mortality of genetic origin, due to a protein deficiency that results in the progressive degeneration of motor neurons, leading to significant weakness and muscle atrophy [1,18]. The severity of SMA depends on the number of functional *SMN2* gene copies, which modulate the harmful effects of *SMN1* deficiency, affecting the age of onset and prognosis [19,20,21].

### 1.2. Polygenic Diseases

Polygenic neurodegenerative diseases are driven by the complex interplay of multiple genes and environmental factors, making their underlying mechanisms more difficult to target with therapeutic interventions. Unlike monogenic disorders, polygenic diseases do not follow a simple inheritance pattern, and their genetic basis often involves variations in several genes, each contributing modestly to disease susceptibility. Alzheimer’s disease (AD), Parkinson’s disease (PD), and amyotrophic lateral sclerosis (ALS) are prime examples of polygenic neurodegenerative diseases, each with a multifactorial etiology involving both genetic predispositions and environmental influences. These disorders were chosen in this review for their high prevalence, significant societal impact, and the progress made in understanding their genetic contributions, despite the challenges posed by their polygenic nature. Recent gene therapy approaches in these diseases aim to modify broader biological processes, such as enhancing cell survival and reducing toxic protein accumulations, rather than targeting a single causative gene [22,23,24,25,26,27,28,29,30,31,32,33,34,35,36,37,38,39,40,41,42].

-Alzheimer’s disease (AD) is the leading cause of dementia [24]. Although the pathogenesis of the sporadic form is unknown, familial AD (<5% of cases) shows mutations in genes coding for amyloid-beta precursor protein (APP), presenilin 1 and 2 (PSEN1-2), and microtubule-associated protein Tau (MAPT). The accumulation of β-amyloid plaques and tangles of hyperphosphorylated tau is associated with memory loss and cognitive decline, with death typically occurring 5–12 years after diagnosis [43].-Parkinson’s disease (PD) is characterized by resting tremor, bradykinesia, and postural instability, along with cognitive impairment [33,44]. Motor dysfunction is produced by the degradation of dopaminergic neurons in the substantia nigra pars compacta [36,45,46]. PD can be sporadic (10–15%) or familial (85–90%). The average age of onset for PD is estimated to be around 60 years, though some patients experience significantly earlier onset. PD is linked to increased mortality and a reduced life expectancy compared to the general population, particularly when diagnosed before the age of 70 [47,48]. Alterations have been identified in the membrane-interacting gene α-synuclein (*SNCA*), [49,50]; in mitochondrial quality control genes, such as phosphatase and tensin homolog (*PTEN*) and leucine-rich repeat kinase 2 (*LRRK2*) [49,51]; in oxidative damage control genes, such as the gene that encodes the parkinsonism associated deglycase (PARK7); and the gene that encodes the lysosomal enzyme glucosylceramidase Beta 1 (GBA1) [52].-Amyotrophic lateral sclerosis (ALS) is a neurodegenerative disorder affecting motor neurons, with both familial (20%) and sporadic (80%) forms [53]. In both, protein aggregation leads to neuronal degeneration, which leads to progressive paralysis and, often, death within 3–5 years [54]. ALS is frequently associated with the mutation of the superoxide dismutase 1 (*SOD1*) gene, causing the accumulation of abnormal SOD1 protein [54], and with the mutation of the TAR DNA-binding protein 43 (*TDP-43*) gene. The TDP-43 protein plays a crucial role in the regulation of RNA splicing, which is essential for the proper expression of genes. Additionally, the dysregulation of TDP-43 has been linked to alterations in neurotransmitter systems, particularly affecting gamma-aminobutyric acid (GABA), an important inhibitory neurotransmitter in the brain. These changes can disrupt neural signaling and may contribute to various NDDs [42,55].

Gene therapy is defined as the process of modifying the structure or expression of genetic material for the treatment of genetically based diseases caused by the mutation of one or a small number of genes [3]. This approach involves delivering therapeutic genetic material through various carriers, such as viral vectors [56], liposomes [57], nanoparticles [58], or exosomes [59]. These systems are designed to transfer genes or oligonucleotides efficiently, while aiming for minimal toxicity.

Ideally, gene therapy should be delivered in a single dose through a safe, minimally invasive route [14,60]. Effective gene therapy for NDDs requires a thorough understanding of disease-related metabolic pathways, the bioavailability of the therapeutic molecules [7,31,46], and the specific properties of neuronal circuits involved [43,50,51].

Gene therapy can be administered through two primary approaches: (1) ex vivo, where genetically modified cells are reintroduced into the patient, and (2) in vivo, where genes or oligonucleotides are delivered directly through naked DNA sequences, nonviral particles, or plasmid or viral vectors [61]. Ex vivo techniques typically involve the use of human pluripotent cells (PSCs), mesenchymal stem cells (MSCs), or hematopoietic stem cells (HSCs) [62]. In vivo techniques include gene addition strategies and genome editing methods, such us CRISPR-Cas9 (clustered regularly interspaced short palindromic repeats associated protein 9), as well as mRNA modifications using antisense oligonucleotides (ASOs) or small-interfering ribonucleic acid (siRNA) for gene silencing. However, these RNA-based therapies have shown limited effectiveness in treating neurological disorders due to depot effects, where over 99% of therapeutic RNA becomes trapped in endosomes. This entrapment significantly hinders the delivery of RNA to target sites. Efforts are underway to develop slow-release RNA therapies capable of sustaining long-term responses, such as those required for PD. These advancements aim to enhance endosomal escape by freeing up 50% or more of the RNA therapeutics, enabling a more rapid and efficient delivery to target cells [63].

Vectors can be packaged into viral and nonviral vehicles. Lentiviruses (LVs) are RNA viruses that have a reverse transcriptase and have been used in preclinical research [34] and clinical trials for NDDs [64]. LVs efficiently target a wide range of tissues and cell types, including the central nervous system. Their ability to stably integrate transgenes into the host genome supports sustained transgene expression. LVs are particularly advantageous for ex vivo gene correction, as these viruses have evolved to preferentially transduce human cells, providing some protection from the immune system. However, the risk of insertional mutagenesis during clinical applications remains a significant concern, and in vivo gene transfer with LVs faces challenges due to immune-mediated rejection [65,66]. In contrast, AAVs and their engineered capsid variants are extensively utilized in clinical settings [8,9,10,12,13,14,67]. AAVs, which belong to the Parvoviridae family, exhibit low oncogenicity and immunogenicity [1,68]. They offer high and stable transduction efficiencies, particularly in fully differentiated neural cells with minimal turnover, and generally present manageable side effects. Among AAVs, AAV9 is particularly favored for NDDs due to its ability to cross the blood–brain barrier (BBB) [16,69]. Nonviral vectors, typically nanoparticles, are used to deliver small genetic sequences and, more rarely, DNA plasmids, which are employed less frequently due to size constraints. These nanoparticles can have organic (e.g., lipid-based, liposomal, or polymeric) or inorganic structures, offering a high degree of design flexibility for targeted tissue delivery [70]. In the context of NDDs, nanoparticles are often administered intracranially, as they generally cannot cross the BBB. However, innovative approaches are emerging that may allow these vectors to bypass this barrier. Despite these advances, the lack of extensive clinical trials leaves unresolved questions regarding their safety and potential side effects [71].

The route administration for gene therapy is determined by several factors, including the desired systemic or localized effect, the specific target cells, the accessibility of these cells (e.g., overcoming barriers, such as the BBB or blood–nerve barrier), the titer and yield of the therapeutic product, and the potential side effects associated with high doses, such as inflammation caused by the administration method. This is crucial for determining the effectiveness and safety of the treatment. The advantages and disadvantages of each route of administration have been discussed in many publications [15,34,60]. Direct intracerebral delivery is preferred for localized effects [32,35,45]. Although it is highly invasive, it is used for the administration of vectors that do not cross the BBB or when high concentrations are needed. On the other hand, intravenous, intra-arterial, or intrathecal routes are less invasive but present challenges related to vector delivery into the central nervous system, as they must be able to cross the BBB [20]. The drawback of those routes of administration is the presence of immune reactions and toxicity [10,13,72]. Currently, the intravenous route has replaced the intra-arterial route. The intrathecal route consists of administering treatment in the subarachnoid space along the neuraxial axis and does not require crossing the BBB. Intracerebroventricular administration consists of direct drug delivery into the ventricles of the brain. This route seems to show the best concentration/invasiveness ratio, achieving a higher effect with lower doses; although, it can cause neurotoxicity [14]. Emerging delivery methods, such as the intranasal [73] and the intrauterine route [15], are under development to improve accessibility and reduce invasiveness.

Current treatments for NDDs primarily focus on symptom management and slowing disease progression [73], but recent advancements in gene therapy have shown promising potential in addressing the underlying genetic causes of these disorders, both in preclinical and early clinical trials [20]. However, the success of gene therapies is contingent upon several key factors, including the safety and efficacy of delivery vectors, the ability to achieve targeted gene expression, and the successful translation of preclinical results into human applications. This review aims to provide a comprehensive analysis of the current state of gene therapy techniques for the most prominent NDDs, including AD, ALS, HD, PD, and SMA. It evaluates their efficacy, safety profiles, and emerging research directions. The outcomes of this review will help identify the current challenges facing gene therapy, particularly the translation from animal models to clinical practice, and highlight future directions.

## 2. Methods

A systematic review was conducted to identify studies investigating gene therapy applications in NDDs, adhering to the PRISMA (Preferred Reporting Items for Systematic Reviews and Meta-Analyses) guidelines [74]. The initial search was conducted on 15 June 2023, with an updated search on 15 December 2023, across multiple databases, including PubMed, Scopus, and Web of Science. The search targeted publications released between January 2020 and December 2023. Search strategies were adapted specifically for the Web of Science (WoS) database, focusing on the topic “gene therapy in neurodegenerative diseases”.

### 2.1. Search Strategy

To optimize search effectiveness and ensure relevant results, the PICO (patient, intervention, comparison, and outcome) framework was employed. Boolean operators OR (to combine free terms) and AND (to connect different search concepts) were utilized to refine the search. Keywords included combinations of terms associated with gene therapy and specific neurodegenerative diseases, such as “neurodegenerative diseases” OR “Alzheimer’s disease” OR “amyotrophic lateral sclerosis” OR “Huntington’s disease” OR “Parkinson’s disease” OR “spinal muscular atrophy” AND “gene therapy”.

The inclusion criteria for this review consisted of full-text, peer-reviewed articles published in English between January 2020 and December 2023. Eligible studies focused on gene therapy interventions for NDDs, including AD, ALS, HD, PD, and SMA. Studies were required to assess the therapeutic efficacy, safety, or clinical outcomes of gene therapies in either clinical or preclinical settings. Both clinical trials and descriptive studies were considered if they evaluated the impact of gene therapy on disease pathophysiology or progression.

Exclusion criteria included studies that were not peer-reviewed, not available in English, or not published in indexed academic journals. Additionally, studies were excluded if they focused on nongenetic therapeutic approaches or addressed diagnostic methods or administration routes without direct evaluation of gene therapy. Publications that did not match the search descriptors or that examined diseases outside the scope of interest were also excluded. Finally, studies published before 2020 were excluded from the review.

### 2.2. Selection Process and Critical Appraisal

The initial search yielded 1465 articles. After removing duplicates and screening titles and abstracts for relevance, 84 articles remained for full-text review. Based on the application of inclusion and exclusion criteria, 26 articles were excluded due to their primary focus on administration methods, diagnostic techniques, or unrelated clinical trials. Finally, 58 articles met the inclusion criteria and were critically appraised by two independent reviewers to ensure methodological quality and relevance. From this pool, 32 studies were selected, comprising preclinical and clinical trials (n = 28) and descriptive studies (n = 4), all of which focused on assessing the effectiveness of gene therapies in neurodegenerative diseases. A flow diagram illustrating the steps taken during the literature review process was created (Figure 1). Each selected study was evaluated independently by the researchers for methodological rigor, with particular attention to research design, statistical analysis, and the reported outcomes. Discrepancies between reviewers were resolved through discussion, ensuring consistency in the selection of high-quality studies for inclusion in the final review. A qualitative synthesis of the results was then prepared to inform the discussion section.

For clarity, summary tables were constructed for each disease studied. These tables presented the authors, year of publication, and the key characteristics of the selected studies, including methodology, country of origin, and participant information. Special emphasis was placed on the curative potential of the gene therapies reviewed.

## 3. Results

The results of this systematic review are based on a final selection of 32 articles, identified through the PRISMA method (Figure 1). These included 17 preclinical studies, 11 clinical trials, and 4 descriptive studies, all focused on evaluating the effectiveness of gene therapy interventions for various NDDs. Our review aimed to assess the therapeutic potential of gene therapies targeting specific NDDs, with a working hypothesis that these therapies could either provide curative effects by addressing underlying genetic mutations or promote cell survival to slow disease progression. The studies reviewed encompass gene therapy approaches for spinal muscular atrophy (n = 8, Table 1), Huntington’s disease (n = 3, Table 2), amyotrophic lateral sclerosis (n = 6, Table 3), Parkinson’s disease (n = 12, Table 4), and Alzheimer’s disease (n = 3, Table 5). The tables below summarize key findings from these studies, including study design, methodology, and reported outcomes, to provide a comprehensive overview of current progress in gene therapy for NDDs.

Based on the review of selected articles, we identified two principal approaches in gene therapy for NDDs, summarized in Figure 2. The first approach is mutation-specific therapies with curative intent, which aim to directly target and correct specific genetic mutations responsible for monogenic NDDs. These therapies involve the introduction of corrective genes or oligonucleotides using AAV vectors or gene-editing technologies, such as CRISPR-Cas9 and antisense oligonucleotides (ASOs). Examples include targeting the HTT gene in Huntington’s disease or the *SMN1* gene in spinal muscular atrophy (Figure 2). These therapies are designed for curative outcomes by addressing the root cause of the disease at the genetic level.

In contrast, the second approach involves cell-survival-promoting therapies with disease-modifying intent, which aim to enhance neuronal survival and slow disease progression without directly correcting the underlying genetic mutation. These therapies utilize adenoviral vectors or stem cell transplantation to deliver neuroprotective factors, inhibit toxic protein aggregation, or modulate gene expression. This broader approach is applicable to both monogenic and polygenic neurodegenerative diseases, offering therapeutic potential for complex diseases, such as Alzheimer’s, Parkinson’s, and amyotrophic lateral sclerosis. This classification helps clarify the distinction between direct mutation-targeting therapies and broader strategies aimed at promoting cell survival and mitigating disease progression (Figure 2). These recent therapies, classified under curative and disease-modifying categories, are detailed in the sections below.

### 3.1. Therapies with Curative Intent

In the case of SMA, onasemnogene abeparvovec (Zolgensma^®^) is an approved gene therapy drug that delivers a functional copy of the *SMN1* gene, which is mutated in SMA, via a viral vector [18]. Clinical trials have demonstrated its efficacy, particularly in motor function, as measured by scales, like CHOP INTEND, when compared to historical cohorts [8,9,12,13]. The therapeutic success is influenced by two key factors: (1) the dose administered and (2) the age and weight at the time of administration. The START:I trial [11], demonstrated that higher doses of Zolgensma^®^ correlate with greater efficacy. In addition, some studies have evaluated the effect of these factors [9,10,12,13,18] and have confirmed that asymptomatic lower-weight patients achieve better motor and survival outcomes. For this reason, a treatment protocol has been established in asymptomatic patients with prenatal diagnosis [10]. While intrauterine administration of the therapy has been explored in animal models, it has not yet proven viable [15].

Years later, nusinersen (Spinraza^®^) was developed, showing significant results in the acquisition of motor milestones for patients with SMA [11]. However, onasemnogene abeparvovec is typically prioritized over nusinersen for SMA treatment, as Zolgensma^®^ requires only a single intravenous dose; whereas, nusinersen involves repeated intrathecal administrations [18,60].

In the case of ALS, tofersen (Qalsody^®^) is an antisense oligonucleotide-based intrathecal therapy designed to treat ALS patients with mutations in the *SOD1* gene. Treatment with tofersen has been shown to decrease the concentration of the SOD1 protein in the cerebrospinal fluid; although, no significant impact on disease progression has been demonstrated [37]. Gene-editing approaches targeting the *SOD1* gene have shown modest outcomes. For instance, gene editing in an ALS mouse model resulted in increased survival, improved muscle atrophy, and reduced SOD1 protein inclusions [39].

Therefore, future research is required to optimize these approaches, and hence, it could be an important line of treatment that may develop further.

Finally, in Huntington’s disease, the administration of Tominersen (developed by ROCHE) has been inconclusive due to the allelic heterogeneity presented in the *HTT* gene mutation [75]. Current research efforts are aimed at blocking mRNA transcription of the *HTT* gene using noncoding RNAs, such as shRNA and miRNA [16]. Although these approaches have yielded promising results in terms of molecular outcomes, they have not yet translated into significant improvements in clinical parameters [6].

### 3.2. Therapies with Modifying Intent

Disease-modifying gene therapies focus on delivering neurotrophic factors or neuroprotective agents through vectors or stem cells, particularly in NDDs lacking a specific genomic target. These therapies are also being explored in disorders with known genetic alterations where targeted interventions have proven insufficient.

In Huntington’s disease, stem-cell-based therapies have produced more significant benefits than strategies using adeno-associated viral vectors [4,7]. Notably, a lower dose of cells seems to be enough to confer neuroprotection, as evidenced by superior outcomes in the lower dose group in a rat model [4].

It has been suggested that PD [36] and AD [22] may share a common therapeutic target involving the neuroprotective functions of astrocytes and microglia. In PD, intracerebral vector administration is the most extensively studied approach in preclinical animal models. Moreover, this strategy has shown better clinical and biochemical outcomes [31,32,35,36,45]. Additionally, DOPA-synthesizing and microglia-restoring stem cells have also provided significant results. However, there is no consensus regarding the optimal administration route, with some studies favoring intracerebral delivery [34], while others report encouraging outcomes from systemic, intravenous administration [28].

In AD, the results have been less favorable. Although gene therapies have enhanced cell survival, they have not consistently restored microglial function or significantly reduced β-amyloid plaque accumulation [24].

Lastly, a potential neuroprotective system based on the homeostatic regulation of autophagy and vesicular trafficking has been discovered in ALS. This system could be modulated either by increasing lysosomal activity [42] or by delivering transport-related proteins, such as synaptotagmin-13 [40]. Nonetheless, research with neuro-protective molecules has achieved better survival outcomes so far [41].

## 4. Discussion

The current treatments based on the gene therapy of five types of NDDs have been addressed, including both therapies that have already been approved and potential therapeutic approaches.

Diseases, from the perspective of gene therapy, can be broadly divided into two categories: those with actionable genetic mutations and those with polygenic inheritance, where no single actionable gene target exists. In NDDs with an actionable mutation, cells harboring driver mutations exhibit sensitivity in response to targeted therapies. For these cases, gene editing or gene therapy techniques, which consist of the introduction or removal of a gene or oligonucleotide, are employed.

However, extrapolating findings from animal models to human patients remains challenging, as only humans possess the complete mutational and pathogenic burden. The utility of any given animal model must be evaluated within the context of the specific research question being addressed. To enhance translational success to clinical applications, therapies should, whenever possible, be tested across multiple preclinical models. This approach is crucial given our limited understanding of NDDs and the fact that no model fully replicates human disease. Moreover, alignment is essential between the timing of observed therapeutic effects in the preclinical model and the stage of pathology in the intended clinical trial population [76].

In diseases without actionable mutations, or in cases where allelic heterogeneity is too extensive to develop patient-specific therapies, disease-modifying treatments are used. These therapies aim to enhance cellular survival and reduce the toxicity derived from protein inclusions or oxidative stress (free radicals). Notably, some NDDs share pathogenic mechanisms, which suggest common therapeutic targets. For instance, both Alzheimer’s and Parkinson’s exhibit microglia dysfunction. Besides, in Huntington’s, ALS and SMA defects in lysosomal transport and cellular autophagy mechanisms contribute to protein aggregation and inclusion. Regulating these processes may help alleviate cellular toxicity and improve outcomes in these diseases [42].

It is worth noting that gene therapy approaches for diseases with more advanced research and preclinical data tend to focus heavily on biodistribution, safety, and pharmacovigilance. For example, extensive studies have been conducted in PD [34] and SMA [10,13,14,60], underscoring the importance of these aspects in therapeutic development.

Regarding the routes of administration, one major challenge in NDDs, unlike other diseases, is overcoming the BBB [1,62,77]. Direct delivery methods, such as intracerebral administration, appear to be the most effective [14]. While subcutaneous administration of neuroprotective factors has shown significant results in PD animal models, these effects have not been replicated in humans, highlighting the superiority of intracerebral routes [44]. However, NDDs do not consist only of local pathological processes, but rather, they tend to spread throughout the central nervous system.

In gene therapy for NDDs, intracerebral administration requires highly precise delivery devices to overcome the BBB and achieve therapeutic efficacy. Stereotactic injection systems are widely employed for this purpose, both in clinical trials and preclinical studies. These systems enable precise delivery of therapeutic agents to specific brain regions but are invasive and require specialized surgical expertise, posing certain risks. For instance, intracerebral injections in adeno-associated virus gene therapy have shown promising improvements in patient symptoms; yet, they are associated with complications, such as intracranial hemorrhage and transient headaches [78]. Thus, stereotactic intracranial approaches can accommodate a variety of gene therapy vehicles, including adeno-associated viral vectors, lentiviral and adenoviral vectors, as well as antisense oligonucleotides and other small molecules. This approach also allows for lower doses of adeno-associated viral vectors, reducing immune responses compared to intravenous or cerebrospinal-fluid-based deliveries [79]. Convection-enhanced delivery is commonly integrated with stereotactic systems, where a catheter is precisely positioned in the brain using stereotactic guidance. A continuous pressure gradient is then applied to facilitate the infusion of therapeutic agents over an extended area of brain tissue, thereby improving distribution and penetration. This approach enhances gene therapy efficacy by promoting more effective delivery of therapeutic agents, such as viral vectors, across larger brain regions [78,80,81,82]. Although effective, this method is invasive and carries risks, such as catheter misplacement and potential tissue damage. In preclinical studies, microinjection pumps and intracerebral microinfusion devices have been used for long-term or localized delivery of gene therapies in animal models. However, their complexity and invasiveness limit their application in human studies. Emerging techniques, such as focused ultrasound with microbubbles, offer a noninvasive alternative by temporarily opening the BBB. This approach facilitates gene delivery without the direct penetration of the brain tissue and is currently in early testing phases for gene therapy applications [83]. Intranasal delivery is another novel, noninvasive method with experimental potential, as it allows vectors to reach the brain along the olfactory pathway [84]. However, its limited efficiency in delivering larger genetic constructs constrains its applicability in gene therapy for NDDs. Despite this, the intranasal administration of neurotrophin in AD [23] or the opening of the BBB using microbubbles and ultrasounds in PD [73] are being explored showing promising results. This latter technique has successfully enabled the intravenous delivery of therapeutic agents, such as the GDNF protein [30] and the bioactive compound “gastrodin” [85], both of which have been shown to enhance dopamine synthesis by astrocytes. Each of these delivery methods has unique implications for precision, patient safety, and therapeutic outcomes, underscoring the need for continued innovation and optimization of gene therapy delivery systems for NDD treatment. Indeed, each delivery route presents its own challenges, and there might be adverse effects derived from the transfer technique used.

Each of these delivery methods has distinct implications for precision, patient safety, and therapeutic outcomes, emphasizing the need for ongoing innovation and optimization of gene therapy delivery systems for treating NDDs. Importantly, each route poses specific challenges, and adverse effects may arise from the delivery technique itself. Recently, there has been a shift toward using adeno-associated virus serotype 9 (AAV9) vectors via intravenous and intraperitoneal administration, which might reduce risks and potential side effects compared to more invasive methods, offering safer alternative gene therapy applications [14,86].

Intravenous and intrathecal administration can cause systemic toxicity, partly due to the hepatotoxicity of commonly used viral vectors and the high doses required for efficacy. This is the case of the nusinersen for the intrathecal treatment of SMA, which requires a minimum of five doses to increase neuronal survival by 10% [11]. Animal model results, particularly in nonhuman primates, are difficult to apply to humans due to differences in administration volume [60]. On the other hand, although the safety of the intrathecal route has been demonstrated in some studies [54], others raised concerns as intrathecal administration worsened the disease progression in ALS rat models [41].

Evaluating and assessing the effectiveness of gene therapies must also include consideration of adverse side effects. Pharmacovigilance studies on SMA gene therapies have revealed a high risk–benefit ratio, since, although they cause an improvement in motor and respiratory function, they are also associated with short-term [10] and long-term side effects due to an increase in the incidence of tumors [19] and skeletal anomalies [72].

Finally, this study has several limitations. While it aimed to comprehensively evaluate the efficacy and safety of gene therapies for the most prominent NDDs, it was not always feasible to investigate the specific reasons behind the ineffectiveness of certain gene therapies. Additionally, important factors, such as gender differences, which play a significant role in the pathophysiology and progression of NDDs, were not addressed in this review. Another key limitation is the lack of in-depth analysis regarding the timing of therapy administration, a critical determinant of therapeutic success in NDDs. Despite these limitations, our review offers valuable insights by systematically categorizing the latest gene therapy approaches, highlighting both curative and disease-modifying strategies and providing a framework for understanding their therapeutic potential.

This work underscores the importance of continuous research and paves the way for future studies to address these gaps, ultimately advancing the translation of gene therapies into clinical practice. These diseases remain without a cure, and specific biomarkers for diagnosing and tracking the progression of most NDDs are still under investigation. While biomarkers are often measured in CSF, this approach has limitations. However, the development of new, high-sensitivity techniques is enabling the detection of biomarkers at very low concentrations in blood or saliva, offering promising alternatives for clinical use [87].

In the future, continuous and collaborative research efforts will be essential to overcome the scientific, ethical, and clinical challenges associated with gene therapy interventions, facilitating their successful translation into clinical practice. Addressing the adverse effects of these therapies remains a critical priority, as they can significantly impact the risk–benefit ratio for patients. Moreover, it is crucial to further refine our understanding of (1) the natural history and key clinical outcomes of various neurodegenerative disorders and (2) the underlying mechanisms driving neurodegeneration. These advancements will be pivotal in optimizing gene therapy strategies and improving therapeutic efficacy.

## 5. Conclusions

In conclusion, this review highlights several key insights into the potential of gene therapy for neurodegenerative diseases (NDDs):-Gene therapy for NDDs offers significant promise for developing interventions that can either slow disease progression or, in some cases, provide curative outcomes;-Therapies with curative intent focus on addressing specific genetic mutations, while disease-modifying therapies aim to enhance cell survival through the administration of neurotrophic factors or neuroprotective agents;-Therapies with curative intent that have shown promising results in terms of clinical significance include the introduction of a gene or oligonucleotide, particularly for SMA and gene editing techniques for ALS. However, these approaches have not yet achieved meaningful clinical results for HD;-Therapies with modifying intent have demonstrated significant clinical results in PD, using both stem cells and the administration of a neurotrophic factor using vectors. Advances have also been made in HD with stem cell therapies and in ALS with the introduction of protective factors through viral vectors or oligonucleotides. However, in AD, the results obtained so far are more limited, highlighting the complexity of targeting polygenic diseases;-Our analysis identified shared pathogenic pathways among certain NDDs, which may present viable targets for gene therapy. This is particularly evident between PD and AD, as well as HD, ALS, and SMA;-According to the characteristics described of each administration route for gene therapy and based on their ability to effectively cross the BBB and target the central nervous system, the optimal routes for gene therapy administration in NDDs appear to be the following: intra-arterial, intravenous with ultrasound facilitation, as well as intrathecal delivery;-Overall, gene therapy offers a promising avenue for both curative and disease-modifying treatments, but further research is necessary to refine these approaches and achieve more widespread clinical success.

## Figures and Tables

**Figure 1 ijms-25-12485-f001:**
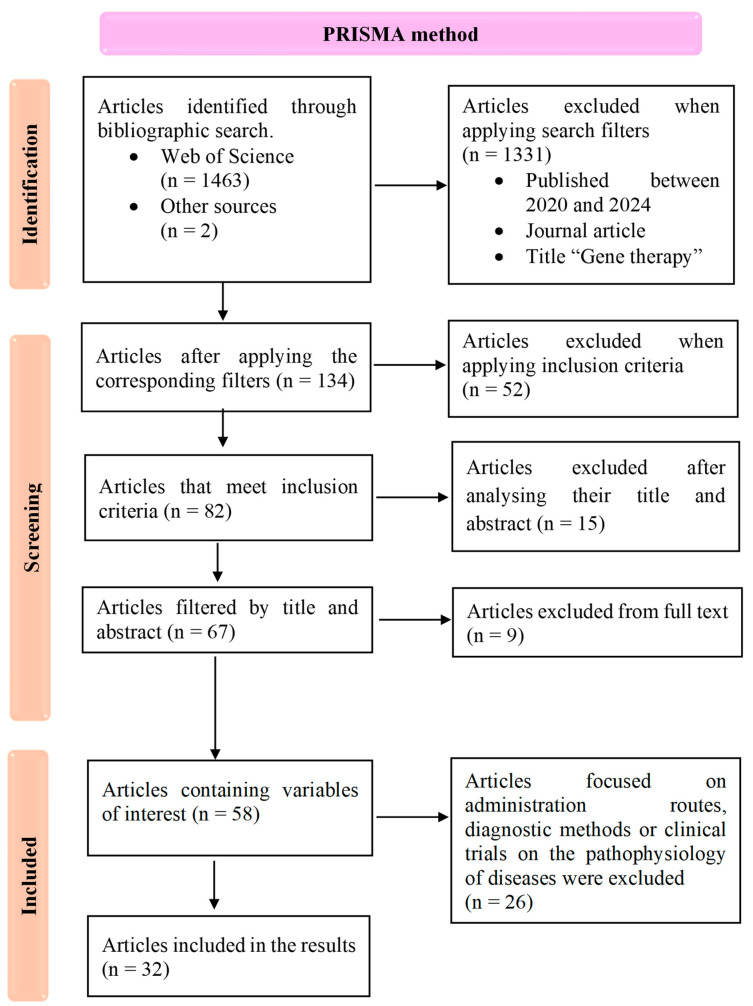
Study selection flowchart following the PRISMA method, detailing the number of studies identified, included, and excluded at each stage, along with the applied exclusion criteria.

**Figure 2 ijms-25-12485-f002:**
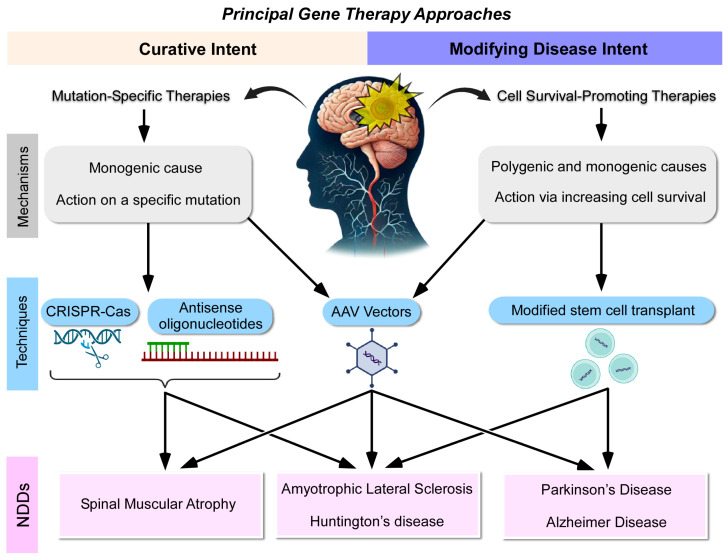
The review analysis highlights two principal gene therapy approaches for neurodegenerative diseases (NDDs). Therapies with curative intent involve the introduction of a corrective gene or oligonucleotide using adeno-associated viral (AAV) vectors or gene-editing techniques (CRISPR-Cas and antisense oligonucleotides), targeting diseases caused by specific genetic mutations. In contrast, therapies with a cell-survival-promoting intent utilize adenoviral vectors or stem cell transplantation to deliver factors that enhance cellular survival. This latter approach can be applied to both monogenic and polygenic neurodegenerative diseases, offering broader therapeutic potential.

**Table 1 ijms-25-12485-t001:** Results: spinal muscular atrophy (SMA).

Reference Year and Country	Clinical/Preclinical	Design	Gene Therapy	Results	Side Effects
Mendell Jerry et al., (2021) [8] USA	Clinical trial; Descriptive study with historical cohort.	Patients with SMA type 1 (n = 15); Age avg: 6.3 months; 5-year follow-up; • Low dose (n = 3) (6.7 × 10^13^ vg/kg); • High dose (n = 10) (1.1 × 10^14^ vg/kg).	IV Onasemnogene Abeparvovec (AAV9-*SMN*).	Significant; Higher achievement of motor development milestones and survival of treatment, when compared to historical cohort. Better results for the high dose group. Long-term favorable safety profile.	Hypersalivation.
Strauss et al., (2022) [9] USA	Clinical trial; Descriptive study with historical cohort.	Presymptomatic patients with SMA type 1 or type 2, with 2 or 3 copies of SMN2 (n = 14); Age avg: 21 days (8–34); 18-month follow up; Dose: 1.1 × 10^14^ vg/kg.	IV Onasemnogene Abeparvovec (AAV9-*SMN*).	Significant; Early administration improves disease progression.	Transient hepatotoxicity, thrombocytopenia, cardiac events, thrombotic microangiopathy, and sensory abnormalities suggestive of ganglionopathy.
Stettner et al., (2023) [10] Switzerland	Clinical trial; Prospective observational study.	Patients with SMA (n = 9); Weight 5.9 ± 1.4 kg; Age avg: 160 days; Dose: 1.1 × 10^14^ vg/kg; • SMA type 1 (n = 6); • SMA type 2 (n = 1); • Presymptomatic (n = 2); Duration 2 years.	IV Onasemnogene Abeparvovec (AAV9-*SMN*).	Significant; Analysis of complications of the therapy.	Hepatotoxicity events; Increased troponin T levels.
Pane et al., (2022) [11] Italy	Case series; Descriptive study with historical Cohort.	Patients with SMA2-3; • SMA type 2: Age: 2.64 to 47.82 months (n = 46); • SMA type 3: Age: 3.21 to 47.82 months (n = 65); Standard nusinersen dosing; Duration: 2.56 years.	Intrathecal nusinersen ASO activator of SMN2 mRNA.	Significant; Improvement in motor function tests in both groups after 12 months, when compared to the historical cohort.	Hepatotoxicity events.
Pane et al., (2023) [12] Italy	Case series.	Patients with SMA type 1 (n = 67) (>6 months, >8.5 kg); Age avg: 20 months; 12-month follow up; Dose: 1.1 × 10^14^ vg/kg.	IV Onasemnogene Abeparvovec (AAV9-*SMN*).	Significant; Asymptomatic patients of younger age and lower age show better results.	Hepatotoxicity events.
Chand et al., (2022) [13] USA	Case series.	Patients with SMA type 1 with two copies of SMN2 (n = 102); • Weight < 8.5 kg (n = 97); • Weight < 13.5 kg (n = 5); Standard onasemnogene abeparvovec dosing.	IV Onasemnogene abeparvovec (AAV9-*SMN*)	Non-significant; Similar safety and presence of adverse events in both groups.	Acute respiratory events, hepatotoxicity, thrombotic microangiopathy, and thrombocytopenia.
Besse et al., (2020) [14] France	Preclinical study.	Transgenic mice; Age: PND1 80 days follow-up; • Control (n = 22); Low dose AAV9: 3 × 10^13^ vg/kg; High dose AAV9: 8 × 10^14^ vg/kg; • IV AAV9-SMN (n = 22); • ICV AAV9-SMN (n = 18).	IV or ICV AAV9-*SMN*.	Significant differences with the control group; No significant differences between IV and ICV in SMN expression; Significant increase in survival in ICV group.	Kyphosis and necrosis in the tail.
Rich et al., (2022) [15] USA	Preclinical study.	Pregnant sows; Gestational age 77 to 110 (corresponding to third trimester in humans) (4–5 piglets) (n = 13); • Treated (n = 11); Doses: 2.0 × 10^13^ vg/kg; 6.5 × 10^12^ vg/kg; • Control (n = 2).	IV shRNA-AAV9.	Non-significant; Not a viable model for the study of in utero viral delivery of gene therapy.	Premature birth.

Abbreviations: AAV9: adeno-associated virus 9; ASO: antisense oligonucleotide; ICV: intracerebroventricular; IV: intravenous; PND1: postnatal day 1; shRNA: short hairpin RNA; SMA: spinal muscular atrophy; SMN: motor neuron survival protein; vg: vector genomes.

**Table 2 ijms-25-12485-t002:** Results: Huntington’s disease (HD).

Reference Year and Country	Clinical/Preclinical	Design	Gene Therapy	Results	Side Effects
Wencesalau et al., (2022) [4] Brazil	Preclinical study.	Male Wistar rats treated with 3-nitropropionic acid; 8 weeks old (n = 60); • Control groups; • Single or three doses of 1 × 10^6^ or 1 × 10^7^ hIDPSCs.	Intraperitoneal administration of hIDPSCs.	Significant; BDNF levels were restored with a low dose of cells; Based on these data, a phase I clinical trial showed that IV administration of hIDPSCs is well tolerated and improves HD motor dysfunction [5]; Phase II clinical trial is under analysis.	No side effects were observed.
Kotowska-Zimmer et al., (2022) [6] Poland	Preclinical study.	YAC128 mice; 12–16 weeks old; • Control group (n = 9); • miRNA group Dose: 1 × 10^11^ gc (n = 10); • shRNA group (n = 10); Dose: 3 × 10^11^ gc.	Intrastratial injection of AAV5-miRNA or AAV5-shRNA.	Significant mHTT reduction in both groups compared to control without affecting endogenous HTT; miRNA reduced spleen weight to values characteristic of wild-type mice and improved motor function.	Abnormal behavior of animals treated with AAV5-shRNAs, and some of them had to be killed before termination of the study; No signs of toxicity were observed in the miRNA group.
Ferlazzo et al., (2023) [7] Italy	Preclinical study.	Mice (R6/2) (n = 44); 4 weeks old; • Control; • CAG repeats; In vitro human neural precursor cells HD.	IV AAV9-*MTF1*.	Significant reduction in mHTT aggregate formation; Amelioration of motor defects.	No side effects were observed.

Abbreviations: AAV5: adeno-associated virus 5; AAV9: adeno-associated virus 9; BDNF: brain-derived neurotrophic factor; HD: Huntington’s disease; hIDPSC: human immature dental pulp stem cells; mHTT: mutant huntingtin protein; IV: intravenous; *MTF1*: metal response element binding transcription factor 1; miRNA: microRNA; shRNA: short hairpin RNA; YAC: yeast artificial chromosome.

**Table 3 ijms-25-12485-t003:** Results: amyotrophic lateral sclerosis (ALS).

Reference Year and Country	Clinical/Preclinical	Design	Gene Therapy	Results	Side Effects
Miller et al., (2020) [37] USA	Phase I/II clinical trial; Descriptive study with historical cohort.	Adult patients with *SOD1* mutations (n = 50); Four dose cohorts; Ascending-dose trial: 20 (n = 10), 40 (n = 9), 60 (n = 9), or 100 mg (n = 10) or placebo (n = 12); 31 weeks follow-up.	Intrathecal administration of tofersen.	Significant decrease in SOD1 concentrations in CSF at the highest concentration of tofersen at day 85; Neurofilament concentrations decreased; Non-significant improvement in clinical parameters.	Lumbar-puncture-related adverse events in most participants; Pleocytosis in some patients; Headache, procedural pain, post–lumbar puncture syndrome, and falls; Two died from pulmonary embolism and respiratory failure.
Mueller et al., (2020) [38] USA	Case report.	Two patients with familial ALS (*SOD1* mutations); Patient 1: 22-year-old; Patient 2: 56-year-old; Dose: 4.2 × 10^14^ vg.	Intrathecal infusion of AAV encoding a microRNA targeting *SOD1*.	Patient 1: Lower SOD1 levels in spinal cord on autopsy; Levels of SOD1 in CSF were only transiently slightly lower; Transient improvement in the strength of his right leg, but no change in vital capacity; Patient 2: stable scores on a composite measure of ALS function and stable vital capacity during a 12-month period.	In Patient 1, meningoradiculitis developed after the infusion; Patient 2 was pretreated with immunosuppressive drugs and did not have this complication.
Lim et al., (2020) [39] USA	Preclinical study.	Male transgenic mice hSOD1 (n = 14); 8 weeks old; Dose: 8 × 10^10^ vg (10 µL).	Intrathecal delivery of AAV9-CRISPR-Cas9.	Significantly delayed progression in the final phase of the disease; Non-significant in the early stages.	No side effects were observed.
Nizzardo et al., (2020) [40] Italy	Preclinical study.	Male and female *SOD1* transgenic mice at 80 days of age (n = 9); Dose: 11 × 10^11^ particles.	Intramuscular injection of AAV9-*SYT13*.	Significant; SYT13 overexpression increases survival and reduces motor neuron pathology.	Not mentioned.
S. Wang et al., (2022) [41] USA	Preclinical study.	Mice (n = 3–18 animals per experiment/group); • Control; • Transgenic *SOD1*; Dose: 2 × 10^13^ gc/mL (10 µL).	Subpial administration of AAV1-*CAV-1*.	Significant; Increase in survival (10%); Delayed disease onset and progression.	Adverse effects from the subpial surgery on body weight at the late stage of disease, with no effect on disease onset or survival.
Tejwani et al., (2023) [42] USA	Preclinical study.	Mice (age P1) divided in four groups (n = 3–28 animals per experiment/group): mutated (*TDP-43*) and nonmutated, treated and nontreated; Dose: 10 µg.	ICV delivery of ASO-Nlk.	Significant; Treatment reduces TDP-43 inclusions and related pathology, increases life span, and ameliorates motor behavior.	Not mentioned.

Abbreviations: AAV1: adeno-associated virus 1; AAV9: adeno-associated virus 9; ASO: antisense nucleotides; *CAV-1:* caveolin-1; CSF: cerebrospinal fluid; ICV: intracerebroventricular; Nlk: nemo-like kinase; SOD1: superoxide dismutase 1; SYT13: synaptogamin 13; TDP-43: transactive response DNA-binding protein 43 kDa; vg: vector genomes.

**Table 4 ijms-25-12485-t004:** Results: Parkinson’s disease (PD).

Reference Year and Country	Clinical/Preclinical	Design	Gene Therapy	Results	Side Effects
Rocco et al., (2022) [25] USA	Phase I clinical trial; Descriptive study with historical cohort.	Patients with advanced PD (n = 13); Age: 51–75 years (mean 65.1 ± 6.4); Three dose cohorts: 450 mL each side; 1.9 × 10^10^ vg (n = 6); 2.3 × 10^11^ vg (n = 6); 9.9 × 10^11^ vg (n = 1); 36–60 months follow-up.	Bilateral putaminal infusions with iMRI aid. AAV2-*GDNF* and gadoteridol.	Administration: Imaging platforms improve therapeutic distribution, increase safety, enable administration customization, and make it easier to measure dosage effectiveness and efficacy.	No evidence of significant, persistent localized trauma or an inflammatory response.
Onuki et al., (2021) [26] Japan	Clinical trial. Descriptive study with historical cohort.	Patients with AADC deficiency (n = 8); Age: 4–18 years (mean 9.5 ± 4.5); 2 × 10^11^ vg each side (200 µL); 2 years s follow-up.	Bilateral putamen AAV2-h *AADC*.	Significant disappearance of painful dystonia; Improvement of motor functions; The cortico-putaminal network is restored.	Not mentioned.
Tai et al., (2022) [27] Taiwan	Phase I/II clinical trial; Descriptive study with historical cohort.	Patients with AADC deficiency (n = 26); Age: 1.7–8.5 years (mean 5.4 ± 2.6); Phase I/II: 1.8 × 10^11^ vg (each side) (n = 10); Phase 2b: 2.4 × 10^11^ vg (each side) (n = 5); 5.4 years follow-up.	Bilateral putamen eladocagene exuparvovec (AAV2-h *AADC*).	Significant de novo dopamine production; Improvement in motor and cognitive function; Improved quality of life.	Potentially related to the surgical procedure; Temporary mild dyskinesia events.
Chen et al., (2020) [28] USA	Preclinical study.	MPTP-induced Neurodegeneration; Mice (n = 3–15 animals per experiment/group); 10–14 weeks old; Dose: 2.0 × 10^6^ cells.	IV transplantation of HSC-GDNF.	Significant improvement of motor and nonmotor symptoms; Restoration of dopamine levels in SNpc.	No side effects were observed.
Gupta et al., (2020) [29] USA	Preclinical study.	Mice (n = 19); Aged 6–12 weeks: • Control; • Injection of C3 or Fluorescent; • 6-OHDA; Dose: ~2 × 10^9^ vg (2 µL).	Administration of AAV2-C3 transferase.	No significant differences between C3 and control groups in behavioral and histological analyses groups.	Transduced dopaminergic neurons express C3 continuously without apparent adverse effects.
Wang et al., (2020) [30] China	Preclinical study.	6-OHDA rats: • Control; • 3 groups treated with ultrasounds; One of them receives gene therapy; Dose: 4 × 10^7^ microbubbles.	Ultrasound combined with lV administration of GDNF retrovirus-loaded microbubbles.	Significant increase in dopamine concentration and number of dopaminergic neurons in the treated group; Improvement of motor symptoms.	Not mentioned.
Cui et al., (2021) [31] USA	Preclinical study.	Transgenic VMAT2 lo mice in two groups (n = 5–6 animals per experiment/group): • 12 months of age; • 18 months of age; Dose: 1 × 10^8^ TU (1–2 µL).	Intracerebral administration of lentivirus mRNA transcription Factors, Phox2a/2b, Hand2, and Gata3.	Significant improvement of motor symptoms and spatial memory by increasing norepinephrine and dopamine concentration.	Not mentioned.
Fernández-Parrilla et al., (2022) [32] Mexico	Preclinical study.	Male rats aged 2 months (n = 120): • Control (n = 20); • 6-OHDA (n = 40); • Treated group (n = 60); Dose: 30 nM; 60 days follow up.	Intracranial nanoparticle-mediated h*CDNF* gene delivery.	Significant re-establishment of the nigrostriatal circuits, motor, and nonmotor deficits.	No side effects were observed.
Kip et al., (2022) [33] USA	Preclinical study.	PD neurodegeneration rat model (6-OHDA) (n = 9); Adult rats; Dose: 1.4 × 10^13^ vg/mL (200 nL).	DBS. AAV5-opsins.	Significant improvement in the motor symptoms of PD.	Selective stimulation of target neurons may reduce side effects compared to electrical DBS.
J. Li et al., (2022) [34] China	Preclinical study.	•Male monkeys, 8–15 years old (MPTP/MPP+ model) (n = 12); • Male rats, 8–10 weeks (6-OHDA model) (n = 19); • Female mice, 6–8 weeks old (n = 8); Dose: 1 × 10^5^ cells/µL.	Intrastriatal administration of MSCs-DOPA.	Significant; Intrastriatal delivery restores dopamine levels and improves motor deficit; Long-term benefit (up to 51 months).	Safety of MSC-DOPA was assessed in mice in detail; No side effects were observed.
Azevedo el al., (2023) [35] Netherlands	Preclinical study.	Female Wistar rats (n = 184), 7 weeks old; • Control (n1 = 20, n2 = 44); • Low dose (n = 32) 4.1 × 10^9^ TU; • High dose (n = 88) 1.0 × 10^10^ TU; Applied either continuously or intermittently.	Intrastriatal administration of doxycycline inducible AAV vector, AAV2/5-V16-h*GDNF*.	Significant; A low-dose or an intermittent high-dose treatment resulted in neuron protection and motor symptom reduction, without inducing deleterious effects.	Continuous administration of high-dose GDNF caused a 50% increased level of oxidized DNA in the substantia nigra pars compacta and aberrant sprouting.
Narváez et al., (2023) [36] USA	Preclinical study.	Male Wistar rats (200–300 g) (6-OHDA) groups (n = 4 animals per experiment/group); Dose: 20,000 cells (3–4 µL).	Intrastriatal CRISP/Cas9-lentivirus; Activation of tyrosine hydroxylase gene.	Significant. Increased dopamine production by astrocytes in the striatum; Transplanted TH-producing astrocytes induced motor recovery.	Not mentioned.

Abbreviations: 6-OHDA: 6-hydroxydopamine; AADC: aromatic L-amino acid decarboxylase; AAV2: adeno-associated virus 2; AAV5: adeno-associated virus 5; DBS: deep brain stimulation; GDNF: glial-cell-derived neurotrophic factor; hCNDF: human cerebral dopamine neurotrophic factor; HSC: hematopoietic stem cells; iMRI: real-time intraoperative magnetic resonance imaging; IV: intravenous; MPTP: 1-methyl-4-phenyl-1,2,3,6-tetrahydropyridine; MSCs: mesenchymal stem cells; SNpc: substantia nigra pars compacta; PD: Parkinson’s disease; TH: tyrosine hydroxylase; TU: transducing units; vg: vector genomes; VMAT2: vesicular monoamine transporter 2.

**Table 5 ijms-25-12485-t005:** Results: Alzheimer’s disease (AD).

Reference Year and Country	Clinical/Preclinical	Design	Gene Therapy	Results	Side Effects
Jeon et al., (2020) [22] South Korea	Preclinical study.	Female Sprague-Dawley rats, 8 weeks old; Dose: 2 × 10^12^ vg/mL (2 µL).	Intrahippocampal injection AAV1-Rheb (S16H).	Significantly increased expression of BDNF in hippocampal neurons but not in astrocytes.	Not mentioned.
Capsoni and Cattaneo et al. (2022) [23] Italy	Preclinical study.	Ts65Dn mice at 4 months of age with accumulation of APP (n = 6); Dose: not mentioned.	Intranasal delivery of hNGFp.	Significant; Treatment rescues astrogliosis, dystrophic microglia, and neurogenesis deficits.	Intranasal delivery reduces the possibility of side effects due to leakage of the protein in blood circulation. hNGFp has similar neurotrophic potency to wildtype NGF, while showing reduced pain sensitization potency.
Mishra et al., (2023) [24] USA	Preclinical study.	5xFAD transgenic mice at 2 months of age (n = 46); Dose: 2 × 10^6^ cells (100 µL).	IV transplantation of HSPCs.	Significant improvement in cognitive function, locomotor activity. Microglia activation and neuroinflammation reduction; Decrease in β-amyloid plaques.	No side effects were observed.

Abbreviations: 5xFAD: 5 mutations familial Alzheimer disease; AAV1: adeno-associated virus 1, APP: amyloid-beta precursor protein; BDNF: brain-derived neurotrophic factor; hNGFp: human nerve growth factor painless; HSPCs: hematopoietic stem and progenitor cells; Rheb (S16H): Ras homolog enriched in brain; Ts65Dn: mouse model of Down’s syndrome that are trisomic for about two-thirds of the genes orthologous to human chromosome 21; vg: vector genomes.

## Data Availability

No new data were created or analyzed in this study. Data sharing is not applicable to this article.

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
