# Peer review of "Advances and Challenges in Gene Therapy for Neurodegenerative Diseases: A Systematic Review"

_ijms, 2024, doi:10.3390/ijms252312485_

Round 1
Reviewer 1 Report
Comments and Suggestions for Authors
L40: Can you please replace death with “reduced life expectancy”
L50-52: Please add references and clinical trial numbers
L80-82: Please add references and clinical trial numbers
L99: Please mention Parkinson’s prognosis/life expectancy
L110: extra full stop
L110: Please mention ALS prognosis/life expectancy
L115: Genetic material is not only transferred via vectors, i.e. siRNA can be delivered naked. Please adjust accordingly
L115: while aiming to minimal toxicity
L117: Effective gene therapy for NDDs requires ….
L124: Direct gene therapy can also be achieved using naked DNA sequences and non-viral particles
L129: Gene therapy also includes gene addition. Also clarify that ASOs and mRNAis are used for silencing
L131: Vectors are not viral or non-viral. Vectors are DNA, usually. Please rephrase to: Vectors can be packaged into viral or non-viral vehicles.
L132: Not all vectors integrate. Please delete this phrase. Lentiviruses do integrade but there are 100 more types of other viral capsids that do not integrate.
L134: There are clinical trials and approved products with Lentis…Please rephrase and include references and clinical trial numbers in the text.
L134-135: Please rephrase to: The development of AAVs and its derivative capsid variants are now extensively used in the clinic (References, Clinical trials). The main reasons that the industry is now investing on AAVs is because they usually have manageable side effects as well as high and stable transduction and expression efficiencies, especially to neural cells that are fully differentiated and have minimal turnover. (Feel free to adjust this info to make it more pleasant to the reader)
L139: Nanoparticles usually carry small sequences, and more rarely DNA plasmids due to size limitations. Overall, this paragraph is structured to give a general understanding, if you want to specifically connect this info with NDDs you need to make this clear. If you want to keep it general, nanoparticles are usually administrated intravenously not ICV.
L142: Route of administration depends on: targeted cells, mechanistic accessibility to the targeted cells (Blood-brain or blood-nerve barriers), titer/yield of the therapeutic product and side effects when high doses are used (ie avoiding AAVs intravenous injection and replacing it with ICV can reduce liver toxicity), inflammation resulting due to administration method (ie intraneural effectively targets nerves but subsequent inflammation and trauma makes it hard to be applied in the clinic)
L148: Intrathecal in not a systemic route, is intra-CSF injection it goes down to the nerves with pressure gradient and a part of it gets in the circulation indirectly. But is not considered a systemic route as into a systemic injection.
L166: to the most prominent examples of CNS NDDs.
L193: Include the dose for all therapies. Discriminate pre-clinical from clinical. Add columns for: side effects and age of participants/animal models
L199: Include doses. Add columns for: side effects and age of participants/animal models
L204, 210, 218: Include the dose for all therapies. Discriminate pre-clinical from clinical. Add columns for: side effects and age of participants/animal models
L323: In discussion also comment about: Treatment responsive biomarkers for NDDs, which is the ideal age for a gene therapy treatment for these patients, what is the disease burden and why is important to consider one-off treatments. Can you quickly comment if alternative treatment methods exist, rather than gene therapy?
Author Response
Thank you very much for taking the time to review this manuscript. We have made the requested modifications, which include the addition of new references to support and strengthen the revised sections. In addition, the tables have been modified to include more information. Moreover, following the editors' instructions, the title and the section order were changed. Consequently, some changes have been applied. Those changes have been highlighted in our submission.
We followed your suggestions, and we think they helped us to improve the quality of our manuscript. We hope you will now find our research suitable for publication.
A point-by point response to each comment is listed below:
REVIEWER 1
- L40: Can you please replace death with “reduced life expectancy”
Answer: The term “death” was changed based on your suggestion.
- L50-52: Please add references and clinical trial numbers
Answer: We have added the references corresponding to the gene therapy studies for HD and SMA selected in this systematic review.
- L80-82: Please add references and clinical trial numbers
Answer: We have added the references corresponding to the gene therapy studies for AD, PD and ALS selected in this systematic review.
- L99: Please mention Parkinson’s prognosis/life expectancy
Answer: We have included a brief mention in this paragraph. “The average age of onset for PD is estimated to be around 60 years, though some patients experience significantly earlier onset. PD is linked to increased mortality and a reduced life expectancy compared to the general population, particularly when diagnosed before the age of 70” (Lines 93-97).
- L110: extra full stop
Answer: Corrected.
- L110: Please mention ALS prognosis/life expectancy
Answer: We have added “which leads to progressive paralysis and, often, death within 3-5 years” (Lines 105-106).
- L115: Genetic material is not only transferred via vectors, i.e. siRNA can be delivered naked. Please adjust accordingly
Answer: Thank you. We agree. We have rephrased this sentence. “This approach involves the administration of delivering therapeutic genetic material through various carriers, such as viral vectors [56], liposomes [57], nanoparticles [58], or exosomes [59]. These systems are designed to transfer genes or oligonucleotides efficiently while aiming to minimal toxicity” (Lines 118-121).
- L115: while aiming to minimal toxicity
Answer: Corrected
- L117: Effective gene therapy for NDDs requires ….
Answer: Corrected
- L124: Direct gene therapy can also be achieved using naked DNA sequences and non-viral particles. L129: Gene therapy also includes gene addition. Also clarify that ASOs and mRNAis are used for silencing
Answer: Thank you for your comments. We have included your suggestions, and we added more information (Lines 129-145).
- L131: Vectors are not viral or non-viral. Vectors are DNA, usually. Please rephrase to: Vectors can be packaged into viral or non-viral vehicles.
Answer: Corrected.
- L132: Not all vectors integrate. Please delete this phrase. Lentiviruses do integrate but there are 100 more types of other viral capsids that do not integrate.
Answer: Deleted.
- L134: There are clinical trials and approved products with Lentis…Please rephrase and include references and clinical trial numbers in the text.
Answer: We agree. We included references. We also included the main advantages and disadvantages of lentiviruses as suggested by another reviewer. “Lentiviruses (LVs) are RNA viruses that have a reverse transcriptase and have been used in preclinical research [34] and clinical trials for NDDs [64]. LVs efficiently target a wide range of tissues and cell types, including the central nervous system. Their ability to stably integrate transgenes into the host genome supports sustained transgene expression. LVs are particularly advantageous for ex vivo gene correction, as these viruses have evolved to preferentially transduce human cells, providing some protection from the immune system. However, the risk of insertional mutagenesis during clinical applications remains a significant concern, and in vivo gene transfer with LVs faces challenges due to immune-mediated rejection [65, 66].” (Lines 149-158).
- L134-135: Please rephrase to: The development of AAVs and its derivative capsid variants are now extensively used in the clinic (References, Clinical trials). The main reasons that the industry is now investing on AAVs is because they usually have manageable side effects as well as high and stable transduction and expression efficiencies, especially to neural cells that are fully differentiated and have minimal turnover. (Feel free to adjust this info to make it more pleasant to the reader). L139: Nanoparticles usually carry small sequences, and more rarely DNA plasmids due to size limitations. Overall, this paragraph is structured to give a general understanding, if you want to specifically connect this info with NDDs you need to make this clear. If you want to keep it general, nanoparticles are usually administrated intravenously not ICV.
Answer: Thank you. We have rephrased this paragraph to include your suggestions and specifically refer this info to NDDs (Lines 158-176).
- L142: Route of administration depends on: targeted cells, mechanistic accessibility to the targeted cells (Blood-brain or blood-nerve barriers), titer/yield of the therapeutic product and side effects when high doses are used (ie avoiding AAVs intravenous injection and replacing it with ICV can reduce liver toxicity), inflammation resulting due to administration method (ie intraneural effectively targets nerves but subsequent inflammation and trauma makes it hard to be applied in the clinic)
Answer: We agree. We have added these factors that depend on the route of administration (Lines 177-182).
- L148: Intrathecal in not a systemic route, is intra-CSF injection it goes down to the nerves with pressure gradient and a part of it gets in the circulation indirectly. But is not considered a systemic route as into a systemic injection.
Answer: We agree. This is a mistake. We have removed the term “systemic”
- L166: to the most prominent examples of CNS NDDs.
Answer: Corrected.
- L193: Include the dose for all therapies. Discriminate pre-clinical from clinical. Add columns for: side effects and age of participants/animal models
L199: Include doses. Add columns for: side effects and age of participants/animal models
L204, 210, 218: Include the dose for all therapies. Discriminate pre-clinical from clinical. Add columns for: side effects and age of participants/animal models
Answer: Thank you. We think these modifications improve the quality of the tables. In Tables, now the clinical trials are shown first, and then the preclinical studies. The title of the “Design Type” column has been changed to “Clinical/Preclinical” to clarify this. We have replaced the column titled “Sample size” with “Design”, to also include the doses and age of the participants/animals. A column for side effects has been included. For some studies, there is some data missing in the publications. We included all the information that is available.
- L323: In discussion also comment about: Treatment responsive biomarkers for NDDs, which is the ideal age for a gene therapy treatment for these patients, what is the disease burden and why is important to consider one-off treatments. Can you quickly comment if alternative treatment methods exist, rather than gene therapy?
Answer: We have included a brief description on these topics at the end of the discussion section. “These diseases remain without a cure, and specific biomarkers for diagnosing and tracking the progression of most NDDs are still under investigation. While biomarkers are often measured in CSF, this approach has limitations. However, the development of new high-sensitivity techniques is enabling the detection of biomarkers at very low concentrations in blood or saliva, offering promising alternatives for clinical use [87]”. (Lines 526-531).

Reviewer 2 Report
Comments and Suggestions for Authors
The article provides a comprehensive review of various gene therapy approaches to treat neurodegenerative diseases. Articles published over a time span of 4 years focussing on relevant parameters such as intervention strategies, routes of delivery and concerns have been reviewed towards this goal. The overall manuscript reads well and authors have been able to convey the main points through a very systematic, concise yet effective manner. Some suggestions to improve the quality are pointed out here and the authors could easily address them. Following these changes, the manuscript can be potentially considered for publication.
- Page 3, line 134- please provide a few more lines to clarify the advantages and disadvantages of lenti viral systems explored. Elaborate a bit more on oncogenic risks for better clarity.
- It would be beneficial to also include an additional statement on lipid nano particles considering its relevance now (page 3, section on delivery systems).
- Intrathecal is not a systemic route as its direct delivery to spinal cord. Line 148 in page 3 is therefore not technically correct.
- Line 150 in page 3: can you provide some more insights into drawbacks of systemic route for CNS delivery specifically? Are there challenges with doses required? Also include relevant literature support to justify additional lines.
- Line 154, page 4: add more details on ICV similar to other routes, for example: it delivers drugs into…, advantages with doses etc?
- Figure 1, page 5: identification title: was it just ‘gene therapy’?
- Line 280 in page 11: gene editing approaches targeting SOD1: please add more details- was this animal study? How long was the study conducted for?
- Lines 285-287, page 11 on HD tominersen- please provide some more background and reference.
- Page 12, last paragraph: it is also worth discussing briefly about the specific modalities/ devices used for intracerebral administration and implications.
- It would also be beneficial to discuss briefly about translational challenges specifically with regard to animal models. It has been mentioned at some instances but readers would be interested to see a little more detailed paragraph on specific challenges.
Author Response
Thank you very much for taking the time to review this manuscript. We have made the requested modifications, which include the addition of new references to support and strengthen the revised sections. In addition, the tables have been modified to include more information. Moreover, following the editors' instructions, the title and the section order were changed. Consequently, some changes have been applied. Those changes have been highlighted in our submission.
We followed your suggestions, and we think they helped us to improve the quality of our manuscript.We hope you will now find our research suitable for publication.
A point-by point response to each comment is listed below:
REVIEWER 2
- Page 3, line 134- please provide a few more lines to clarify the advantages and disadvantages of lenti viral systems explored. Elaborate a bit more on oncogenic risks for better clarity.
Answer: Corrected. We have elaborated briefly more information on this topic. “Lentiviruses (LVs) are RNA viruses that have a reverse transcriptase and have been used in preclinical research [34] and clinical trials for NDDs [64]. LVs efficiently target a wide range of tissues and cell types, including the central nervous system. Their ability to stably integrate transgenes into the host genome supports sustained transgene expression. LVs are particularly advantageous for ex vivo gene correction, as these viruses have evolved to preferentially transduce human cells, providing some protection from the immune system. However, the risk of insertional mutagenesis during clinical applications remains a significant concern, and in vivo gene transfer with LVs faces challenges due to immune-mediated rejection [65, 66].” (Lines 149-158).
- It would be beneficial to also include an additional statement on lipid nano particles considering its relevance now (page 3, section on delivery systems).
Answer: We agree. We have included a statement on the relevance of nanoparticles. “Non-viral vectors, typically nanoparticles, are used to deliver small genetic sequences, and more rarely DNA plasmids, employed less frequently due to size constraints. These nanoparticles can have organic (e.g., lipid-based, liposomal, or polymeric) or inorganic structures, offer a high degree of design flexibility for targeted tissue delivery [70]. In the context of NDDs, nanoparticles are often administered intracranially, as they generally cannot cross the BBB. However, innovative approaches are emerging that may allow these vectors to bypass this barrier. Despite these advances, the lack of extensive clinical trials leaves unresolved questions regarding their safety and potential side effects [71]. (Lines 166-175).
- Intrathecal is not a systemic route as its direct delivery to spinal cord. Line 148 in page 3 is therefore not technically correct.
Answer: We agree. This is a mistake. We have removed the term “systemic”
- Line 150 in page 3: can you provide some more insights into drawbacks of systemic route for CNS delivery specifically? Are there challenges with doses required? Also include relevant literature support to justify additional lines.
Answer: The specific difficulty for delivery to the CNS is the crossing the blood-brain barrier as shown in the text. We have made the requested modifications (red paragraphs), which include the addition of new references to support and strengthen the revised sections.
- Line 154, page 4: add more details on ICV similar to other routes, for example: it delivers drugs into…, advantages with doses etc?
Answer: We agree. We have briefly mentioned this “Intra-cerebroventricular administration consists of direct drug delivery into the ventricles of the brain. This route seems to show the best concentration/invasiveness ratio, achieving a higher effect with lower doses, although it can cause neurotoxicity [14]”. (Lines 194-197).
- Figure 1, page 5: identification title: was it just ‘gene therapy’?
Answer: Yes, this is correct. It was “Gene therapy”.
- Line 280 in page 11: gene editing approaches targeting SOD1: please add more details- was this animal study? How long was the study conducted for?
Answer: We have clarified that this is an animal study and included additional details. The study was conducted until the animals reached the end of their lifespan, demonstrating prolonged survival. In this mouse model, the first signs of the disease typically appear around 94 days of age, and the animals usually exhibited the disorder symptoms by 126 days of age on average.
- Lines 285-287, page 11 on HD tominersen- please provide some more background and reference.
Answer: Thank you. Corrected. We added the reference of the study.
- Page 12, last paragraph: it is also worth discussing briefly about the specific modalities/ devices used for intracerebral administration and implications.
Answer: We agree, and we have included a brief discussion on the specific modalities and devices used for intracerebral administration and their implications for gene therapy (Lines 441-488).
- It would also be beneficial to discuss briefly about translational challenges specifically with regard to animal models. It has been mentioned at some instances but readers would be interested to see a little more detailed paragraph on specific challenges.
Answer: We have included a brief paragraph on this topic in the discussion. “The utility of any given animal model must be evaluated within the context of the specific research question being addressed. To enhance translational success to clinical applications, therapies should, whenever possible, be tested across multiple preclinical models. This approach is crucial given our limited understanding of NDDs and the fact that no model fully replicates human disease. Moreover, alignment is essential between the timing of observed therapeutic effects in the preclinical model and the stage of pathology in the intended clinical trial population [76]” (Lines 410-416).
